# RNA-Seq Profiling of Neutrophil-Derived Microvesicles in Alzheimer’s Disease Patients Identifies a miRNA Signature That May Impact Blood–Brain Barrier Integrity

**DOI:** 10.3390/ijms23115913

**Published:** 2022-05-25

**Authors:** Irina Vázquez-Villaseñor, Cynthia I. Smith, Yung J. R. Thang, Paul R. Heath, Stephen B. Wharton, Daniel J. Blackburn, Victoria C. Ridger, Julie E. Simpson

**Affiliations:** 1Sheffield Institute for Translational Neuroscience, The University of Sheffield, Sheffield S10 2HQ, UK; isabel_9691@yahoo.com (C.I.S.); yjrthang1@sheffield.ac.uk (Y.J.R.T.); p.heath@sheffield.ac.uk (P.R.H.); s.wharton@sheffield.ac.uk (S.B.W.); d.blackburn@sheffield.ac.uk (D.J.B.); 2Department of Infection, Immunity and Cardiovascular Disease, Medical School, The University of Sheffield, Beech Hill Road, Sheffield S10 2RX, UK; v.c.ridger@sheffield.ac.uk

**Keywords:** Alzheimer’s disease, systemic infection, neutrophil, neutrophil-derived microvesicles, miRNA

## Abstract

(1) Background: Systemic infection is associated with increased neuroinflammation and accelerated cognitive decline in AD patients. Activated neutrophils produce neutrophil-derived microvesicles (NMV), which are internalised by human brain microvascular endothelial cells and increase their permeability in vitro, suggesting that NMV play a role in blood–brain barrier (BBB) integrity during infection. The current study investigated whether microRNA content of NMV from AD patients is significantly different compared to healthy controls and could impact cerebrovascular integrity. (2) Methods: Neutrophils isolated from peripheral blood samples of five AD and five healthy control donors without systemic infection were stimulated to produce NMV. MicroRNAs isolated from NMV were analysed by RNA-Seq, and online bioinformatic tools were used to identify significantly differentially expressed microRNAs in the NMV. Target and pathway analyses were performed to predict the impact of the candidate microRNAs on vascular integrity. (3) Results: There was no significant difference in either the number of neutrophils (*p* = 0.309) or the number of NMV (*p* = 0.3434) isolated from AD donors compared to control. However, 158 microRNAs were significantly dysregulated in AD NMV compared to controls, some of which were associated with BBB dysfunction, including miR-210, miR-20b-5p and miR-126-5p. Pathway analysis revealed numerous significantly affected pathways involved in regulating vascular integrity, including the TGFβ and PDGFB pathways, as well as Hippo, IL-2 and DNA damage signalling. (4) Conclusions: NMV from AD patients contain miRNAs that may alter the integrity of the BBB and represent a novel neutrophil-mediated mechanism for BBB dysfunction in AD and the accelerated cognitive decline seen as a result of a systemic infection.

## 1. Introduction

Systemic infection is a risk factor for cognitive decline [1] and accelerates disease progression in Alzheimer’s disease (AD) patients [2]. Recent post-mortem studies have shown terminal systemic infection exacerbates cerebral hypoperfusion and blood–brain barrier (BBB) dysfunction in AD, modifying the neuroinflammatory response [3]. However, the specific mechanism(s) by which the systemic inflammatory response modulates the integrity of the cerebrovasculature remain unknown.

While neutrophil extravasation in the brain is not a common feature of AD pathology, several lines of evidence support the role of these white blood cells in the progression of the disease. Levels of circulating neutrophils have been shown to be significantly higher in AD patients compared to controls [4,5,6] and express higher levels of proinflammatory transcripts [7]. Furthermore, peripheral circulating neutrophil-related inflammatory proteins predict a decline in executive function in patients with mild AD [8], and depletion of neutrophils in a mouse model of AD has been shown to attenuate pathology and improve cognition [9].

Activated neutrophils produce neutrophil-derived microvesicles (NMV), which mediate the immune response under pathophysiological conditions and contribute to vascular inflammation [10]. Different studies have provided evidence of circulating extracellular vesicles being implicated in AD that have been mainly studied as biomarkers of disease progression [11,12], but we recently showed that NMV specifically are internalised by human brain microvascular endothelial cells and increase their permeability in vitro [13], suggesting that NMV may play a role in modulating the integrity of the BBB during an infection. These small (0.1–1 µm diameter) vesicles contain a range of biologically active molecules, including nucleic acids such as microRNA (miRNA) [14]. MicroRNAs are small, non-coding, single-stranded RNAs approximately 22 nucleotides long that regulate gene expression by targeting mRNA transcripts. Several miRNAs have been identified that play a role in the onset and progression of AD pathology; these miRNAs participate in APP processing, amyloid-β metabolism and amyloid-β mediated toxicity, tau synthesis and phosphorylation, neuroinflammation and apoptotic pathways [15]. miRNAs have also been linked to dysfunction of the BBB by altering tight junction proteins, such as ZO-1, claudin-5 and occludin [16,17,18], and their modulation has shown to decrease BBB permeability in vivo and in vitro [18,19]. Therefore, the current study aimed to develop a robust protocol to characterise the miRNA content of NMV in AD patients without systemic infection and identify potential candidate miRNAs that may impact the integrity of the BBB.

## 2. Results

### 2.1. No Significant Difference in the Number of Neutrophils and NMV between AD and Control without Systemic Infection

Neutrophils were isolated from whole blood from five control and five AD donors free of systemic infection at the time of collection. Before ex vivo stimulation, the number of neutrophils isolated per ml of blood was determined. Quantification showed no significant difference between AD and controls in the number of isolated neutrophils (*p* = 0.3095). After stimulation with fMLP, the number of purified NMV per ml of blood did not differ between groups either (*p* = 0.3434) (Figure 1).

### 2.2. The miR Signature of NMV from AD Patients

#### 2.2.1. Quality Control of Small RNA-seq Data

Overall, the quality of the sequencing data was good and suitable for downstream analysis. A low percentage of Per base N calls (>5%) and a sequencing accuracy of 99.9% (phred scores > 30 across reads) were reported. No indication of adapter content was found, signifying successful adapter trimming. According to sequence length distribution, most reads peaked at 20–21 bp, consistent with the miRNA enriched libraries used for this study. Quality warnings and fails for the Per Tile Sequence Quality, Per base sequence content, and GC content, could reflect highly biased sequence compositions; however, small/miRNA libraries are known to produce particularly biased base sequence content [20]; based on this, it was considered that fails were unlikely to affect the downstream analysis of the sequencing data.

#### 2.2.2. Detection of Dysregulated miRNA

A clear separation of the dementia and control groups was shown by Galaxy DESeq 2 (Figure 2), although different levels of variance between groups were shown by the different tools used for the differential expression analysis. The lists of dysregulated miRNA obtained from Galaxy DESeq 2, Degust, OASIS 2.0 and sRNAToolbox were combined according to *p*-value (≤0.05) and log2FC to identify overlapping up and downregulated miRNA and a list of 158 dysregulated miRNA (117 upregulated and 41 downregulated) was generated (Appendix A). Furthermore, the top five dysregulated miRNA from each tool were combined, and after the removal of duplicates, a final list of 15 dysregulated miRNA found across all tools was generated (Table 1).

#### 2.2.3. Target and Pathway Analyses

Over 17,000 mRNA targets from the total 117 upregulated miRNA and over 15,000 mRNA targets from the total 41 downregulated miRNA were identified through target analysis. Using Enrichr, the top 10 most affected pathways found by KEGG (2021) and Bioplanet (2019) are shown in Table 2.

## 3. Discussion

Circulating levels of neutrophils with significantly higher levels of hyperactivation positively correlate with cognitive decline in AD [6,7]. However, in contrast to transgenic mouse models of AD, which display neutrophil accumulation surrounding Aβ plaques [21], there is limited evidence of extensive neutrophil trafficking into the brain in AD. Moreover, neuroinflammation is a hallmark of AD [22,23], and activation of the different stress response mechanisms and inflammatory cascades could also be exacerbated by systemic inflammation. While the mechanism underlying the crosstalk between neutrophils and the CNS remains elusive, the current transcriptomic profiling study reveals that the NMV from AD patients (without systemic infection) contain miRNA, which may target genes associated with maintaining the integrity of the BBB. Therefore, NMV-induced cerebrovascular permeability may represent one potential mechanism whereby neutrophils during a systemic inflammatory response increase the permeability of the BBB, leading to increased neuroinflammation and cognitive decline in AD.

In contrast to previous findings of significantly higher levels of neutrophils in AD patients compared to controls [4], the current study did not detect significant differences in either circulating neutrophils or levels of NMV, likely reflecting age-related differences in neutrophil levels in AD, as well as the small number of cases in this pilot study, highlighting the importance of conducting this study in a larger cohort. However, RNA-seq did reveal significant differences in the miRNA signature of the patient NMV, several of which are known to play a key role in AD and regulate the integrity of the BBB [24].

From the top 15 dysregulated miRNA, miR-4449, miR-151a, miR-502, miR-652-5p, miR-16-1-3p, miR-136-3p, miR-20b-5p and miR-Let-7g-3p have been implicated in ageing, AD and other neurodegenerative diseases. For instance, miR-4449 is significantly downregulated in superior and middle temporal gyrus extracts of AD cases [25], and even though the current study showed upregulation of this miRNA, its expression could vary depending on the tissue under study. Upregulation of miR-151a circulating levels is also observed in AD and ischaemic stroke patients [26,27] and has a role in the strengthening of the endothelial cell barrier in primary lung endothelial cells [28], suggesting it could also impact the BBB. Interestingly, miR-151a circulating levels vary with age [29] as well as with disease stage in amyotrophic lateral sclerosis (ALS) [30] and Parkinson’s disease (PD) [31], where mir-151a levels are downregulated at moderate and severe disease stages; studies in transgenic mouse models of AD have also shown age-dependent expression of miR-652-5p [32], and of miR-16-1-3p [33]. This evidence highlights the importance of age and disease stage in miRNA expression and the relevance of acquiring longitudinal data to see changes over disease progression when investigating NMV miRNA content in AD patients with and without systemic infection.

Previous miRNA profiling studies have identified decreased expression of miR-Let-7g-3p in the blood of AD patients [34,35,36,37], conflicting with the current study that found both arms significantly upregulated in the NMV of AD patients, but total circulating miRNAs could differ from the NMV miRNA cargo after neutrophil stimulation, emphasising the importance of differentiating the effects of circulating miRNA and miRNA from internalised NVM in AD and BBB dysfunction. Overexpression of miR-Let-7g-3p has a neuroprotective effect, including reducing the permeability of the BBB and improving functional outcomes in a mouse model of stroke [38,39], demonstrating that not all miRNA within NMV negatively impact the integrity of the BBB. Other miRNAs that have been suggested as possible disease biomarkers and that were detected in AD NMV include miR-502-3p, which is altered in individuals with AD [26], small vessel vascular dementia [40] and frontotemporal dementia [41], and exosomal miR-136-3p, dysregulated in PD and AD cerebrospinal fluid (CSF) samples [42]. Even though these miRNAs have already been studied in the context of AD, our study is the first one to show they are also part of AD NMV cargo.

miR-20b-5p, one of the top upregulated miRNA, and miR-424-5p, also upregulated in the NMV of AD patients, decrease the expression of tight junction proteins and are implicated in diabetic retinal vascular dysfunction [43], and the increased permeability of brain microvascular endothelial cells [18], respectively. Furthermore, inhibition of miR-424-5p induces expression of the tight junction protein ZO-1 and attenuates neurological dysfunction in a mouse model of stroke [44].

A detailed literature search of other significantly dysregulated miRNAs in AD NMV (Appendix A) suggested their involvement not only in AD but also in BBB integrity. Over-expression of miR-126-3p/-5p decreases the expression of vascular cell adhesion molecules and reduces the production of proinflammatory cytokines [45], while decreased expression is associated with dysfunction of the BBB in a rat model of intracerebral haemorrhage [46]. The current study demonstrates decreased expression of miR-126-5p in the AD NMV, which supports the reduction of this miRNA in peripheral blood of AD patients [35], and suggests the reduction in the expression of miR-126-5p may exacerbate BBB damage in AD.

miR-210, implicated in regulating vascular integrity, is downregulated in AD and inversely correlates with disease severity [47]. Indeed, exosomal delivery of miR-210 has been proposed as a treatment post-stroke [48]. While these studies suggest a neuroprotective role for this miRNA, others have shown that miR-210 is associated with an increase in infarct volume and the behavioural deficits seed post-stroke and suggest that antagonism of miR-210 is a potential therapy [49,50]. Whether the upregulation of miR-210 in the current study plays a beneficial or detrimental role is unknown and requires further investigation.

While studies of animal models suggest a potential role for some candidate miRNAs identified in this study in impacting BBB integrity in AD, it should be acknowledged that there are functional differences across species, which are important considerations in the interpretation of the findings [16]. Furthermore, we have previously characterised the miRNA profile of the BBB in an ageing human and mouse cohort and found that only 15 miRNA overlapped between species [51], highlighting the caution that should be taken when extrapolating the data from rodents. Another important consideration of miRNA profiling is arm information. Mature arms of miRNA may have different targets [52] and opposing effects [53]. For example, miR-21-3p decreases the expression of tight junction proteins and increases vascular leakage in a rat model of traumatic brain injury [54], while miR-21-5p (downregulated in the current study) decreases leakage and neurological deficits [55,56].

Target analysis of the differentially expressed miRNA identified the TGFβ and PDGFβ signalling pathways. TGFβ is a pleiotropic cytokine implicated in several neurodegenerative diseases, including AD [57]; however, its precise role in disease pathogenesis is debated. While some studies suggest increased TGFβ signalling reduces AD pathology [58], others suggest inhibiting TGFβ restores the integrity of the BBB [59] and improves cognition in an ageing mouse model [60]. These conflicting results may indicate cell-specific roles for TGFβ [61]; for example, TGFβ signalling in vascular smooth muscle cells has an anti-inflammatory effect, while it has a proinflammatory effect on endothelial cells [62]. Furthermore, TGFβ signalling is biphasic, with acute activation of the pathway playing a neuroprotective role while chronic activation is neurotoxic [63,64]. PDGFβ signalling also has a role in BBB integrity. PDGFβ is one of the four members of the platelet-derived growth factor family (PDGF), and it is secreted by endothelial cells; it binds to receptors (PDGFRβ) present on pericytes and has been shown to promote BBB integrity through pericyte–endothelium interactions [65,66]. PDGFβ is critical for BBB recovery after cerebral ischaemia as it induces pericyte recruitment, migration and proliferation [67,68,69]. A more recent study has also shown regulation of tight junction protein expression via PDGFβ and TGFβ signalling interactions [70]; moreover, levels of PDGFβ are altered in CSF and plasma from AD patients [71], and AD transgenic mouse models have also shown that a dysfunctional PDGFβ/PDGFRβ axis could lead to pericyte degeneration and BBB breakage [72]. These data suggest that AD NMV miRNA cargo could modulate both pathways and impact the BBB, but further investigation is required to characterise this impact.

Hippo signalling, interleukin-2 (IL-2) and DNA damage pathways could also be involved in BBB integrity maintenance and the response to endothelial cell damage. The Hippo/YAP cascade is necessary for the proliferation and normal function of vascular endothelial cells and participates in their inflammatory response [73,74,75]. Additionally, activation of Hippo signalling proteins YAP and TAZ has a protective effect on the BBB in a model of middle cerebral artery occlusion [76], and the pathway interacts with other signalling cascades, including the Wnt pathway [77,78], also a target identified in the current study and linked to BBB maintenance [79]. IL-2 is a pleiotropic cytokine known to promote brain endothelial cell activation resulting in an inflammatory response, alteration of adherens junctions and BBB disruption [80], and dysregulation of miRNA targeting DNA damage pathways could result in either a positive or an impaired repair response in BBB cells. miRNA regulation of these pathways, together with other antioxidant strategies linked to vitagene network regulation [23,81,82], opens up the possibility of therapeutic modulation to counteract detrimental neuroinflammatory and apoptotic responses and prevent further BBB dysfunction.

We have previously shown that NMV are internalised and modulate mRNA expression in brain microvascular endothelial cells in vitro, resulting in a significant reduction in transendothelial electrical resistance and increased permeability [13]. The current study extends these findings and demonstrates that the NMV from AD patients without systemic infection contain miRNAs that may alter the integrity of the BBB and represent a novel neutrophil-mediated mechanism for BBB dysfunction in AD. Future studies are required to confirm these findings in a larger cohort and to determine whether NMV content is altered in AD patients undergoing a systemic inflammatory response after infection and whether this is involved in the accelerated progression of the disease.

## 4. Materials and Methods

### 4.1. Neutrophil-Derived Microvesicle Isolation

Prior to sample collection, donors were reviewed by a clinician and had a normal temperature and no signs of recent or ongoing infection; C reactive protein levels (CRP) were also measured (Table 3). Peripheral blood neutrophils isolation from whole blood samples and neutrophil stimulation for NMV purification and NMV quantification were performed based on the protocol of Gomez et al., 2020 [10]. Briefly, 40 mL of peripheral venous blood was collected in tubes containing 3.8% sodium citrate; the platelet-rich plasma (PRP) was separated from the cellular component by centrifugation. Neutrophils were isolated by density gradient centrifugation using Histopaque-1077 (Cambridge Bioscience, Cambridge, UK), followed by a lysis of red blood cells. To stimulate neutrophil microvesicle (NMV) formation, isolated neutrophils were incubated with the bacterial-derived peptide fMLP (10 µmol/L, Sigma, Gillingham, UK) for 1 h at 37 °C in 5% CO_2_. Neutrophils and large cell debris were pelleted by centrifugation (500× *g* for 5 min followed by 1500× *g*, 5 min), and the supernatant was centrifuged at 20,000× *g* for 30 min to pellet NMV. The pelleted NMV were resuspended in 0.2 µm sterile-filtered PBS. The number of NMV from each isolation was determined using a BD LSRII flow cytometer (Beckton Dickinson, Wokingham, UK). The settings were standardised on forward-scatter and side-scatter gates using Megamix beads of various sizes (0.5, 0.9 and 3 µm, BioCytex, Marseille, France). NMV were quantified using SpheroTM AccuCount beads (Saxon Europe, Kelso, UK). The flow cytometer was set to count 1000 beads, and the concentration of NMV in the sample was calculated using the manufacturer’s instructions. Statistical analyses were performed in GraphPad Prism 9.0.2 for Windows (GraphPad Software, San Diego, CA, USA); differences between groups in the number of neutrophils and number of NMV per ml blood were determined using a Mann–Whitney test (two-tailed).

### 4.2. Small RNA Purification, Library Preparation and Sequencing

Small RNA content from NMV obtained from healthy control and Alzheimer’s disease donors was extracted using the miRNeasy mini kit (QIAGEN, Manchester, UK) according to manufacturer instructions. Briefly, NMV were disrupted with QIAzol Lysis Reagent, and chloroform was added to the samples for RNA recovery by phase separation. The recovered aqueous phase containing the RNA was mixed with 70% ethanol and transferred to an RNeasy Mini spin column for centrifugation, and the flow-through containing the enriched miRNA fraction was collected. Purification of the enriched miRNA fraction was performed with the RNeasy MinElute Cleanup Kit (QIAGEN). For this, the flow-through recovered in the previous step was mixed with 100% ethanol and centrifuged in an RNeasy MinElute spin column. The column was washed with the RPE buffer, followed by a wash with 80% ethanol and an extra centrifugation step to remove any remaining ethanol. The miRNA-enriched fraction was eluted in 14 µL of RNAse-free water. The quality and quantity of the purified mirRNA-enriched fraction obtained from NMV were assessed with a Bioanalyser Small RNA assay (Agilent Technologies, Cheadle, UK) before cDNA library preparation. To assess the efficiency of small RNA purification, an additional control sample was spiked-in with the C. elegans mir-39 spike-in control, which was then reversed, transcribed and amplified using the miScript RT kit (Qiagen) and the miScript SYBR Green PCR kit, respectively.

The Somagenics RealSeq^®^-Biofluids Plasma/Serum miRNA Library Kit for Illumina^®^ sequencing (Somagenics, Sta Cruz, CA, USA) was used to prepare the cDNA library from the NMV-derived miRNA-enriched fractions. The libraries were prepared following the manufacturer’s instructions and with an initial input of 2 ng of small RNA in a total volume of 9.5 µL per sample. For PCR amplification, the samples were amplified for 20 cycles of PCR (denaturation at 94 °C, 30 s; amplification for 20 cycles at 94 °C 15 s, 62 °C 30 s, 70 °C 15 s and, finally, 70 °C for 5 min). Size selection of the libraries was performed using the SPRIselect^®^ Reagent (SPRI) as indicated by the Somagenics Kit, and libraries were checked for quality and quantified with a Bioanalyser High Sensitivity DNA assay (Agilent Technologies) and a Qubit Fluorometer. For cluster generation, 4 nM of each cDNA library from 10 samples (5 healthy controls and 5 AD) were pooled together and loaded onto two lanes of the flow cell to maximise the number of mapped reads. Samples were processed using the Illumina HiSeq Rapid SBS kit (Illumina, San Diego, CA, USA).

### 4.3. Sequencing Data Analysis

Quality control of sequencing data was done using Galaxy. Sequence data were converted from image.bcl files to raw de-multiplexed sequencing data in fastq text file format. Raw sequencing data were uploaded to Galaxy and processed with the “fastp” tool to trim off the Illumina^®^ TruSeq adapter sequences. Quality control was performed on the individual fastp files using the “FastQC” tool, and the reports were combined with “MultiQC”. The “fastp” files were aligned to the reference genome “Human Dec. 2013 (GRch38/hg38); both the NCBI RefSeq and the sno/miRNA tracks were selected on the “UCSC main table browser” and “htseq-count” was used to obtain counts from the aligned files and the genome tracks.

The count data were analysed using four different online softwares, Galaxy, Degust, Oasis2.0 and sRNAToolbox, to produce a combined list of dysregulated miRNAs:

GALAXY [83]: Differential expression analysis was performed with DESeq2 [84] on the htseq-count files from both genome tracks (NCBI RefSeq and sno/miRNA), which were sorted into “Healthy controls” and “Dementia” groups. Results were filtered to select the miRNA present and sorted to select miRNA with *p* ≤ 0.05.

DEGUST [85]: The htseq-count data generated with Galaxy for both the NCBI RefSeq and sno/miRNA genome tracks were used to create the count matrix that was uploaded to Degust. The minimum read count was set to 10, and files were grouped as “Dementia” and “Controls”. The output file was filtered for miRNA with *p* ≤ 0.05.

OASIS2.0 [86]: Fastq files were uploaded to Oasis2.0 for adapter trimming, quality control analysis and small RNA detection. The resulting files were sorted into either “Control” or “Treatment” groups, and the output was filtered for miRNA with *p* ≤ 0.05.

sRNAtoolbox [87]: Fastq files were uploaded to sRNAbench for adapter trimming and alignment. The resulting files were uploaded to sRNAde, which uses different tools for the differential expression analysis (DESeq [88], DESeq 2 [84], NOIseq [89], EdgeR [90] and t-test); groups were defined as “Control#Dementia”. Results obtained from each tool were filtered for miRNA with *p* ≤ 0.05 or a probability of ≥0.8 (NOIseq). The filtered results were combined in one file, and duplicates were removed.

DE results from each analysis were transferred to one single Excel file, and duplicates were removed either by using the “Remove duplicates” function or by manually removing overlapping miRNA when these were named with a different miRNA method. The final list of 158 DE miRNA across tools was separated according to whether they were over or under-expressed.

### 4.4. Target and Pathway Analyses

miRWalk, miRDB, mIRTarBase and miRSystem were used to identify possible targets of the significantly dysregulated miRNA. For this, the lists of under- and over-expressed miRNA were input into each tool. To combine these results, columns containing the target list from all four tools were transferred to a new Excel file and duplicates were removed. The lists of target genes from the under- and over-expressed miRNA were uploaded to Enrichr for pathways analysis using the Bioplanet 2019 and KEGG 2019 Human tools. Results were sorted based on *p*-value, from lowest to highest, and the top 10 significantly dysregulated pathways were selected.

## Figures and Tables

**Figure 1 ijms-23-05913-f001:**
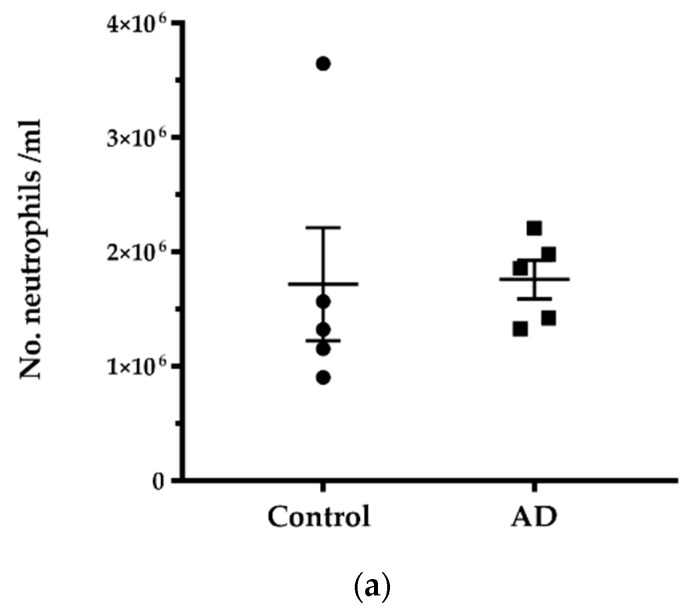
Number of neutrophils and neutrophil microvesicles (NMV) in blood samples isolated from control and AD donors. (**a**) The total number of neutrophils/mL of blood was not significantly different between the two groups. (**b**) The total number of NMV/mL blood purified after neutrophil stimulation ex vivo did not differ significantly between control and AD donors. Mann–Whitney test (two-tailed), *n* = 5 per group.

**Figure 2 ijms-23-05913-f002:**
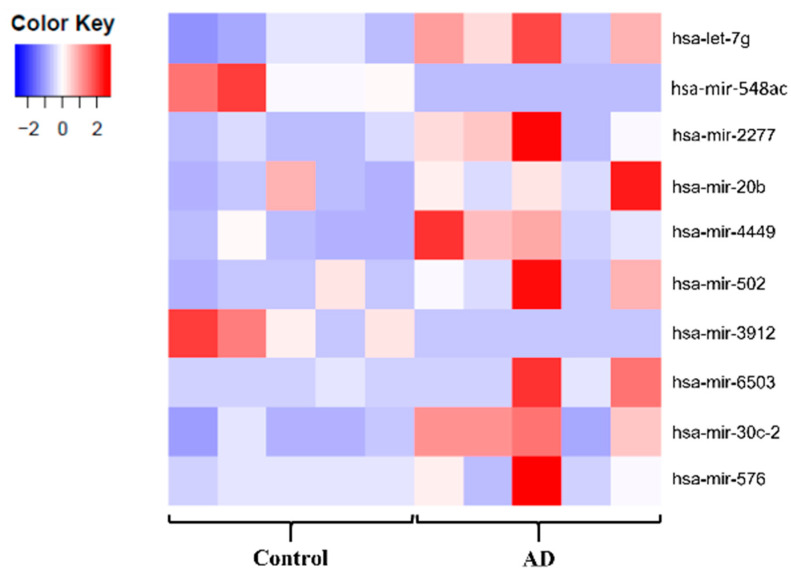
Representative heatmap portraying the top 10 significantly dysregulated miRNAs based on the DESeq 2 (sno/miRNA track) analysis. A clear difference is seen between the AD and control groups.

**Table 1 ijms-23-05913-t001:** Differentially expressed (DE) miRNA found in peripheral NMV from AD subjects based on analysis conducted across different tools.

Tool	Top 5 DE miRNA	*p*-Value	log2FC	Top 15 DE miRNA across Tools	
				Up	Down
**DESeq2—sno/miRNA track**	MIRLET7G	0.00002	1.36		
	MIR548AC	0.00106	−2.22		
	MIR20B	0.0014	1.58		
	MIR502	0.0016	1.56		
	MIR4449	0.00329	1.32		
**DESeq2—NCBI RefSeq track**	hsa-let-7g	0.00037	1.2		
	hsa-mir-548ac	0.00055	−1.9		
	hsa-mir-2277	0.00262	1.65	hsa-let-7a-3p	hsa-mir-151a
	hsa-mir-20b	0.00415	1.31	hsa-let-7g	hsa-mir-548ac
	hsa-mir-4449	0.00534	1.19	miR-4449	hsa-miR-4485-3p
**Degust—** **sno/miRNA track**	hsa-mir-20b	0.00062	1.98	miR-652-5p	hsa-miR-136-3p
	hsa-let-7g	0.00139	1.37	miR-16-1-3p	hsa-miR-584-5p
	hsa-mir-151a	0.00199	−1.66	miR-20b-5p	
	hsa-mir-4449	0.00373	1.47	miR-502	
	hsa-mir-6503	0.0073	1.74	miR-2277	
**Degust—NCBI RefSeq track**	microRNA 20b	0.0003	2.05	miR-4443	
	microRNA let-7g	0.00032	1.43	miR-6503	
	microRNA 4449	0.00151	1.57		
	microRNA 151a	0.00277	−1.51		
	microRNA 6503	0.00391	1.7		
**OASIS 2.0**	hsa-miR-4485-3p	2 × 10^−7^	−1.19		
	hsa-miR-652-5p	0.000001	2.82		
	hsa-let-7a-3p	0.000004	2.34		
	hsa-miR-4443	0.00002	2		
	hsa-miR-16-1-3p	0.000027	3.52		
**sRNAToolbox**	hsa-miR-136-3p	0.000074	−1.49		
	hsa-miR-584-5p	0.0028	−1.02		
	hsa-miR-4485-3p	0.01	−1.75		
	hsa-let-7a-3p	0.02	3.14		
	hsa-miR-20b-5p	0.02	2.35		

**Table 2 ijms-23-05913-t002:** Pathways altered by dysregulated NMV miRNA targets in AD individuals.

	Upregulated miRNA Targets			Downregulated miRNA Targets		
	Term	*p*-Value	Adjusted *p*-Value	Term	*p*-Value	Adjusted *p*-Value
**Bioplanet 2019**	Axon guidance	3.67 × 10^−11^	4.22 × 10^−8^	Generic transcription pathway	1.04 × 10^−11^	1.57 × 10^−8^
	Developmental biology	5.59 × 10^−11^	4.22 × 10^−8^	Wnt signalling pathway	8.47 × 10^−11^	5.69 × 10^−8^
	Interleukin-2 signalling pathway	2.31 × 10^−8^	9.24 × 10^−6^	Developmental biology	1.13 × 10^−10^	5.69 × 10^−8^
	Insulin signalling pathway	2.50 × 10^−8^	9.24 × 10^−6^	Pathways in cancer	2.02 × 10^−10^	7.61 × 10^−8^
	TGF-beta regulation of extracellular matrix	3.28 × 10^−8^	9.24 × 10^−6^	Interleukin-2 signalling pathway	5.63 × 10^−10^	1.70 × 10^−7^
	Wnt signalling pathway	4.22 × 10^−8^	9.24 × 10^−6^	Axon guidance	4.35 × 10^−9^	1.10 × 10^−6^
	Generic transcription pathway	4.28 × 10^−8^	9.24 × 10^−6^	PDGFB signalling pathway	5.66 × 10^−9^	1.22 × 10^−6^
	Pathways in cancer	7.50 × 10^−7^	1.39 × 10^−4^	TGF-beta regulation of extracellular matrix	2.32 × 10^−8^	4.22 × 10^−6^
	PDGFB signalling pathway	8.27 × 10^−7^	1.39 × 10^−4^	Pancreatic cancer	2.51 × 10^−8^	4.22 × 10^−6^
	TGF-beta signalling pathway	3.39 × 10^−6^	4.85 × 10^−4^	ATM-dependent DNA damage response	3.03 × 10^−8^	4.57 × 10^−6^
**KEGG 2021 Human**	Proteoglycans in cancer	5.13 × 10^−7^	9.85 × 10^−5^	Endocytosis	1.32 × 10^−9^	4.21 × 10^−7^
	Axon guidance	6.16 × 10^−7^	9.85 × 10^−5^	Pathways in cancer	1.09 × 10^−8^	1.75 × 10^−6^
	T cell receptor signalling pathway	1.26 × 10^−5^	1.19 × 10^−3^	AGE-RAGE signalling pathway in diabetic complications	5.69 × 10^−8^	6.07 × 10^−6^
	AGE-RAGE signalling pathway in diabetic complications	1.94 × 10^−5^	1.19 × 10^−3^	MAPK signalling pathway	1.62 × 10^−7^	1.30 × 10^−5^
	Focal adhesion	2.15 × 10^−5^	1.19 × 10^−3^	Signalling pathways regulating pluripotency of stem cells	2.42 × 10^−7^	1.55 × 10^−5^
	Hippo signalling pathway	2.60 × 10^−5^	1.19 × 10^−3^	Proteoglycans in cancer	3.36 × 10^−7^	1.79 × 10^−5^
	MAPK signalling pathway	2.61 × 10^−5^	1.19 × 10^−3^	Herpes simplex virus 1 infection	9.16 × 10^−7^	4.19 × 10^−5^
	Insulin signalling pathway	4.85 × 10^−5^	1.57 × 10^−3^	Pancreatic cancer	1.42 × 10^−6^	5.68 × 10^−5^
	Yersinia infection	4.85 × 10^−5^	1.57 × 10^−3^	Inositol phosphate metabolism	2.78 × 10^−6^	8.99 × 10^−5^
	Cellular senescence	4.92 × 10^−5^	1.57 × 10^−3^	Yersinia infection	2.81 × 10^−6^	8.99 × 10^−5^

**Table 3 ijms-23-05913-t003:** Information for control and AD donors. CRP—C-reactive protein; AD—Alzheimer’s disease; NA—not available.

Group	Age	Sex	Temperature (°C)	CRP (mg/L)
Control	56	F	36.8	NA
	55	F	36.5	1.3
	67	F	36.2	1.0
	71	F	36.7	4.4
	66	M	37.1	0.7
AD	63	F	36.7	8.2
	62	M	36.9	1.2
	53	M	35.8	0.6
	70	M	35.4	0.4
	56	M	36.1	0.3
	61.9 ± 6.6		36.4 ± 0.53	1.1 ± 2.6

## Data Availability

In this section, please provide details regarding where data supporting reported results can be found, including links to publicly archived datasets analysed or generated during the study.

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
