# Peer review of "RNA-Seq Profiling of Neutrophil-Derived Microvesicles in Alzheimer’s Disease Patients Identifies a miRNA Signature That May Impact Blood–Brain Barrier Integrity"

_ijms, 2022, doi:10.3390/ijms23115913_

Round 1
Reviewer 1 Report
Irina et al. have investigated microRNA profile of neutrophil-derived microvesicles in Alzheimer's disease patients using RNA-seq analysis. The authors hypothesized that the profile of microRNA may play a role in increasing the ability of them permeating through blood-brain barrier. In order to test the hypothesis, they did microRNA profiling using the RNAseq and compared healthy patients and AD patients. They concluded that there are few microRNAs that are upregulated or downregulated that may play role in affecting the integrity of BBB integrity.
The manuscript can not accepted in its current form. Following changes are warranted.
1) The manuscript require some additional introduction. For example, they should include more details about role of microRNAs in general for AD or inflammation. Also, they should include the significance of looking into NMV in detail and if any current literature exist and what gaps are trying to fill.
2) The authors have cited previous study where the authors did observe in increase in neutrophills, in contrast, the authors did not observe that. The authors should include more details about why there is different observation. The authors have described age-related difference in neutrophills.
3) The authors should include if they have verified or looked into the expression of these microRNAs using PCR or looked into genes from the pathways that are affected from the elevation of these microRNAs.
Author Response
Response to Reviewer 1
We thank the reviewer for their comments. We have addressed these as follows:
Point 1.A. The manuscript requires some additional introduction. For example, they should include more details about role of microRNAs in general for AD or inflammation.
Response to Point 1.A We have expanded the introduction where we say, "Several miRNAs have been identified which play a role in the onset and progression of AD pathology [13], and dysfunction of the BBB [14]". The revised paragraph reads as follows: “Several miRNAs have been identified that play a role in the onset and progression of AD pathology; these miRNAs participate in APP processing, amyloid-β metabolism and amyloid-β mediated toxicity, tau synthesis and phosphorylation, neuroinflammation and apoptotic pathways [13]. miRNAs have also been linked to dysfunction of the BBB by altering tight junctions proteins, such as ZO-1, claudin-5 and occludin [14–16], and their modulation has shown to decrease BBB permeability in transgenic mouse and in vitro models [16,17].”
Point 1.B. Also, they should include the significance of looking into NMV in detail and if any current literature exists and what gaps are trying to fill.
Response to Point 1.B. To highlight the significance of investigating NMV we referred to our previous study (Ajikumar et al., 2019) in the introduction, where we say “We recently showed that NMV are internalised by human brain microvascular endothelial cells and increase their permeability in vitro [11], suggesting that NMV may play a role in modulating the integrity of the BBB during an infection.”
To the best of our knowledge, there are no other studies which have looked at the interaction between NMV and the BBB, but we have mentioned in the introduction studies which have looked at extracellular vesicles in AD as disease biomarkers. The revised paragraph reads as follows: “Activated neutrophils produce neutrophil-derived microvesicles (NMV) which mediate the immune response under pathophysiological conditions and contribute to vascular inflammation [10]. Different studies have provided evidence of circulating extracellular vesicles being implicated in AD and that have been mainly studied as biomarkers of disease progression [11,12] but we recently showed that NMV specifically are internalised by human brain microvascular endothelial cells and increase their permeability in vitro [13], suggesting that NMV may play a role in modulating the integrity of the BBB during an infection.”
Point 2. The authors have cited previous study where the authors did observe in increase in neutrophils, in contrast, the authors did not observe that. The authors should include more details about why there is different observation. The authors have described age-related difference in neutrophils.
Response to Point 2. This short communication paper presents data obtained from a pilot study which was performed with a small number of cases; we have acknowledged this in the discussion and have also mentioned that this could be the reason why we did not observe a difference in neutrophil numbers, compared to other studies (Discussion, 2nd paragraph: “In contrast to previous findings of significantly higher levels of neutrophils in AD patients compared to controls [4], the current study did not detect significant differences in either circulating neutrophils or levels of NMV, likely reflecting age-related differences in neutrophil levels in AD, as well as the small number of cases in this pilot study, highlighting the importance of conducting this study in a larger cohort. However, RNA-seq did reveal significant differences in the miRNA signature of the patient NMV, several of which are known to play a key role in AD and regulating integrity of the BBB [22].)
Even though we mentioned this could also be linked to age-related differences between our cohort and other studies, we are cautious as to not overinterpret the data and speculate about other reasons for not finding a difference in neutrophil numbers based on such a small cohort and have added a sentence indicating the importance of performing this study in a larger cohort to confirm our findings regarding neutrophil levels in AD patients and controls.
Point 3. The authors should include if they have verified or investigated the expression of these microRNAs using PCR or looked into genes from the pathways that are affected from the elevation of these microRNAs.
Response to Point 3. We agree with the reviewer that additional validation studies are required. Our aim is to publish this preliminary data as a short communications paper with significant results that warrant further investigation in the future.
Reviewer 2 Report
Alzheimer's disease (AD) is an incurable neurodegenerative disease diagnosed by clinicians through healthcare records and neuroimaging techniques. These methods lack sensitivity and specificity, so new antemortem non-invasive strategies to diagnose AD are needed. Moreover, systemic infection is a risk factor for cognitive decline and accelerates disease progression in Alzheimer’s disease (AD) patients. Recent post-mortem studies have shown terminal systemic infection exacerbates cerebral hypoperfusion and blood-brain barrier (BBB) dysfunction in AD, modifying the neuroinflammatory response. However, the specific mechanism(s) by which the systemic inflammatory response modulates the integrity of the cerebrovasculature remain unknownTarget analysis of the differentially expressed miRNA identified Hippo signalling, interleukin-2 (IL-2), and DNA damage pathways could also be involved in BBB integrity maintenance and the response to endothelial cell damage.
Interplay and coordination of redox interactions with endogenous and exogenous antioxidant defence systems is an emerging area of reserach interest in anticancer and antidegenerative therapeutics. Moreover, particular attention has been given to providing an assessment of the quantitative features of the dose-response relationships and underlying mechanisms that could account for the biphasic nature of the hormetic response after exposure to redox active agents, such as free radical oxygen species and their impact in inflammatory/antinflammatory pathways. The hormetic dose response should be seen as a reliable feature of the dose response for oxygen free radicals and their redox regulated transcriptional factors as well as antioxidant compounds and appears to have an important impact on brain pathophysiology and stress resistance mechanisms to oxidative and inflammatory insult and neurodegenerative damage.
This is an interesting paper. The study is well-conceived and well-executed. This reviewer is satisfied with the significance of this study, the care in which the study was performed, and the implications of the results for human health. Results presented are convincing, the work does not raises any foundamental concern which need to be addressed. The questions posed are of extremely high interest,the paper does give adequate definitive information, therefore pending some minor point which Authors may consider to address is possible to accept for publication.
Minor concerns:
- Preconditioning signal leading to cellular protection through Hormesis is an important redox dependent aging-associated to free radicals species accumulation, inflammatory responses involved in the pathophysiology of AD. This aspect should be highlighted in the discussion and references properly added (Calabrese et al., Nature Neurosci., 2007 8, 766; Calabrese et al., 2010, Antiox. Redox Signal 13,1763; Trovato et al., Immun Ageing. 2018 Feb 14;15:8)
- Given the relationship between vitagene network and its possible biological relevance in resilience mechanisms underlying pathophysiology of AD, Authors while interpetrating results can illustrate appropriately this aspect and make proper connection in the discussion. These references can be quoted (Mancuso C., et al., (2006) Redox Rep. 11, 207-213; Miquel S, et. Al., Ageing Res Rev. 2018 42:40-55; Drake J., et al., (2003) Journal of Neuroscience Research 74, 6, 917 - 927).
Author Response
Response to Reviewer 2:
This is an interesting paper. The study is well-conceived and well-executed.
We thank the reviewer for their positive comments and support of the study.
Point 1. Preconditioning signal leading to cellular protection through Hormesis is an important redox dependent aging-associated to free radicals species accumulation, inflammatory responses involved in the pathophysiology of AD. This aspect should be highlighted in the discussion and references properly added (Calabrese et al., Nature Neurosci., 2007 8, 766; Calabrese et al., 2010, Antiox. Redox Signal 13,1763; Trovato et al., Immun Ageing. 2018 Feb 14;15:8)
Response to Point 1. We have added this to the discussion. The revised paragraph reads as follows (Discussion, 1st paragraph): “Moreover, neuroinflammation is a hallmark of AD [22,23], and activation of the different stress response mechanisms and inflammatory cascades could also be exacerbated by systemic inflammation.”
Point 2. Given the relationship between vitagene network and its possible biological relevance in resilience mechanisms underlying pathophysiology of AD, Authors while interpetrating results can illustrate appropriately this aspect and make proper connection in the discussion. These references can be quoted (Mancuso C., et al., (2006) Redox Rep. 11, 207-213; Miquel S, et. Al., Ageing Res Rev. 2018 42:40-55; Drake J., et al., (2003) Journal of Neuroscience Research 74, 6, 917 - 927).
Response to point 2. We have added this to the discussion. The revised version reads as follows: “IL-2 is a pleiotropic cytokine known to promote brain endothelial cells activation resulting in an inflammatory response, alteration of adherens junctions and BBB disruption [80], and dysregulation of miRNA targeting DNA damage pathways could result in either a positive or an impaired repair response in BBB cells. miRNA regulation of these pathways, together with other antioxidant strategies linked to vitagene network regulation [23,81,82], opens up the possibility of therapeutic modulation to counteract detrimental neuroinflammatory and apoptotic responses and prevent further BBB dysfunction.”